# Understanding the Development of Heterocyclic Aromatic Amines in Fried Bacon and in the Remaining Oil after Pan-Frying in Five Different Vegetable Oils

**DOI:** 10.3390/foods11213491

**Published:** 2022-11-03

**Authors:** Hongzhen Du, Ziyi Wang, Yuexin Li, Qian Liu, Qian Chen, Baohua Kong

**Affiliations:** College of Food Science, Northeast Agricultural University, Harbin 150030, China

**Keywords:** heterocyclic aromatic amine, bacon, vegetable oil, precursor compound

## Abstract

The physicochemical properties of five vegetable oils (peanut, corn, rapeseed, sunflower seed, and soybean) and their impact on the development of heterocyclic aromatic amines (HAAs) in pan-fried bacon and in the remaining oil were investigated. Corn oil led to the lowest total free amino acids (FAAs) contents and glucose content of fried bacon (*p* < 0.05) and rapeseed oil led to the lowest creatine content of fried bacon (*p* < 0.05). Bacon fried in corn oil had the highest HAA contents (*p* < 0.05). The total HAA contents of the oils after frying were lowest in rapeseed and soybean oils (*p* < 0.05). The type of vegetable oil used affected the color of the fried bacon but not the flavor and taste (*p* < 0.05). To reduce the HAA concentrations of fried bacon, the type of vegetable oil should be considered.

## 1. Introduction

Consumers appreciate bacon for its distinctive texture, taste, and flavor [1]. Bacon is typically prepared from pork bellies, mainly by curing, smoking, and frying; frying is one of the most important stages [2]. The method of frying determines the flavor and texture of bacon. Many kinds of vegetable oil are available for frying foods. Vegetable oils vary in fatty acid composition, color, and flavor. Although vegetable oil imparts flavor and crispiness to meat during frying, it may also promote the development of heterocyclic aromatic amines (HAAs) in the meat [3].

HAAs are food-derived carcinogens and have been substantially studied in meat products and model systems [4]. HAAs were first found in charred fish [5]. In general, the precursors in meat and meat products, such as free amino acids (FAAs), carbohydrates, and creatine, react at high temperatures to form HAAs [6]. To date, about 30 HAAs have been identified in meat or other protein products [7]. Among them, 2-amino-3-methylimidazo[4,5-f]quinoline (IQ), 2-amino-3,4-dimethylimidazo[4,5-f]quinoline (MeIQ), 2-amino-3,8-dimethylimidazo[4,5-f]quinoxaline (MeIQx), 2-amino-3,4,8-trimethylimidazo[4,5-f]quinoxaline (4,8-DiMeIQx), and 2-amino-1-methyl-6-phenylimidazo[4,5-b]pyridine (PhIP) are classified as human carcinogens by the International Agency for Research on Cancer [8]. Many researchers have conducted extensive research on the factors that affect the concentration of HAAs in meat products, including the muscle type [9], the cooking method [10], heating time and cooked level [11], heating method and temperature [2], sugar concentration [12], lipid oxidation [4], amino acids [13], pH value [14], and antioxidants and reducing agents [15]. Although many studies have reported on the HAA contents in meat products, few studies have examined the influence of different vegetable oils on the formation of HAAs during frying. Ekiz et al. [16] found that seven different frying oils (sunflower, hazelnut, canola, commercially mixed, corn, riviera olive, and natural extra-virgin olive) had a noticeable impact on the formation of MeIQx in meatballs. They proposed that the content of unsaturated fatty acids in vegetable oils may be responsible for the differences in HAA contents of the products. Li et al. [17] investigated the effect of 10 different vegetable oils (soybean, rapeseed, peanut, corn, olive, sunflower, rice germ, walnut, torreya seed, and grapeseed) on the development of HAAs in beef patties and reported that the role of vegetable oils in suppression and enhancement of HAA formation are complex and inconsistent.

Only limited publications are available on the relationship between the physicochemical characteristics of vegetable oils and the formation of HAAs in meat products. Furthermore, peanut, corn, rapeseed, sunflower seed, and soybean oils are popularly edible oils in local markets, especially in Northeast China, and are commonly used as frying oils in food processing. This research investigated the effect of five different vegetable oils (peanut, corn, rapeseed, sunflower seed, and soybean) on the development of HAAs in fried bacon as well as in the remaining oil. The level of FAAs, creatine, and glucose and the sensory qualities were also assessed.

## 2. Materials and Methods

### 2.1. Materials and Chemicals

The frying oils (peanut, corn, rapeseed, sunflower seed, and soybean) were acquired from local markets and all of them were refined. The HAA standards, including IQ, MeIQ, MeIQx, 4,8-DiMeIQx, PhIP, 1-methyl-9H-pyrido[3,4-b]indole (Harman), 2-amino-3-methyl-imidazo[4,5-f]-quinoxaline (IQx), 2-amino-9H-pyrido[2,3-b]indole (AαC), 9H-pyrido[3,4-b]indole (Norharman), 2-amino-3-methyl-9H-pyrido[2,3-b]indole (MeAαC), 2-amino-5-phenylpyridine (Phe-P-I), and 2-amino-3,7,8-trimethyl-imidazo[4,5-f]-quinoxaline (7,8-DiMeIQx) and the internal standard 2-amino-3,4,7,8-tetramethylimidazo[4,5-f]-quinoxaline (TriMeIQx) were purchased from Toronto Research Chemicals (Downsview, ON, Canada). These standards were of mass spectral grade (purity >98%). Waters Corporation (Shanghai, China) provided the Oasis MCX cartridges (3 cm^3^, 60 mg). Ethanol, methanol, acetonitrile, n-hexane, dichloromethane, and ethyl acetate (mass spectral grade) were purchased from Merck KGaA (Darmstadt, Germany). The amino acid mixture standard was acquired from ANPEL Laboratory Technologies (Shanghai) Inc. (Shanghai, China). Supelco 37 Component FAME Mix and ammonium acetate were acquired from Sigma-Aldrich (Shanghai) Trading Co., Ltd. (Shanghai, China).

### 2.2. Bacon Preparation

Nine fresh pork belly samples were purchased from Gaojin Meat Corp. (Harbin, China). Each pork belly was cut into quarters and thirty-six pieces (belly chops) were obtained from nine pork bellies. The bacon was prepared based on our previous methods [18]. Brine containing sodium chloride (90 g/kg) and sodium nitrite (500 mg/kg) was injected into the bellies with a pump uptake of 10% using an injector. Then, the bellies were immersed in the above same brine at 4 °C for 20 h. The cured belly chops were divided into three groups. Each group had twelve belly chops, which were treated as a batch and processed independently. Twelve belly chops were a group and treated as a batch and independently processed. The bacon process proceeded as follows: (1) dried at 45 °C/30 min, (2) smoked at 55 °C/1 h, (3) smoked at 65 °C/2 h. After that, the bacon was cooled and sliced into 2 mm thickness. More than 48 slices were obtained from each smoked belly chop. The slices of bacon were placed in plastic package bags. The oxygen transmission capacity and water vapor transmission capacity of packages (Weiguang Plastic Co., Ltd., Lianyungang, China) were less than 0.024 cm^3^/(m^2^·24 h·MPa) and 0.006 g/(m^2^·24 h) at 23 °C, respectively. Each vacuum-packaged sample contained eight slices of bacon and was frozen at −25 °C for at least 20 h to ensure that the core temperature of the bacon was below −18 °C by air freezing. Then, frozen samples were stored in a −18 °C refrigerator until use. The samples were thawed in a 4 °C refrigerator until their temperature was 4 ± 0.5 °C and used within 48 h. Each belly chop acquired six vacuum packages, and 72 packages of bacon (12 belly chops) per batch were obtained. Then, they were randomly divided into six treatments (*n* = 12 packages of bacon per treatment): unfried bacon (control) and bacon fried in peanut oil, corn oil, rapeseed oil, sunflower seed oil, and soybean oil, respectively. Vegetable oil (20 mL) was added to the non-stick pan (32 cm in diameter) and heated for 3–4 min until the pan bottom reached 150–170 °C. Eight slices of bacon were added to the pan and fried for 1 min on each side. After frying, for each treatment, three packages of bacon were used for the HAA contents analysis, four packages of bacon were used for the physicochemical properties analysis, and five packages of bacon were used for sensory evaluation. For each frying treatment, the vegetable oil remaining after frying the bacon was collected and analyzed for HAAs.

### 2.3. Determination of Physicochemical Characteristics of Vegetable Oils

#### 2.3.1. Fatty Acid Composition

The methyl esterification pretreatment of fatty acids in each vegetable oil was performed according to the gas chromatographic method ISO 12966-2 [19]. The vegetable oil (1 g) was mixed with 8 mL of sodium hydroxide (NaOH) 2% methanol solution in a flask. Then, the flask was connected to a reflux condenser and heated at 80 ± 1 °C until the oil disappeared. Subsequently, 7 mL of boron trifluoride 15% methanol solution was added to the reflux condenser, and the flask was heated for a further 2 min at 80 ± 1 °C, and then it was cooled to room temperature. After that, n-heptane (30 mL) and saturated NaCl aqueous solution (10 mL) were added successively to the flask and stood for 20 min. The upper n-heptane extract (5 mL) was mixed with anhydrous sodium sulfate (3 g) in a glass tube and vortexed for 2 min. Finally, the upper solution (1 mL) was filtered and used for further analysis. The contents of fatty acid methyl esters were determined based on the methods of our previous work [20]. The data were presented as the percentage (%) of the corresponding peak area of each fatty acid to the total peak area of all peaks. 

#### 2.3.2. Total Free Fatty Acid (FFA) Content

The FFA content was determined according to the titrimetric method of American Oil Chemists’ Society (AOCS) official methods [21]. The oil sample was mixed with ethanol (50 mL, 95%) and phenolphthalein solution (3 drops). The mixture was titrated with NaOH (0.1 M) solution until the appearance of a permanent pink color (lasting 30 s without fading). The FFA content was calculated according to the equation:FFA (%) = ((V × c × 282)/(1000 × m))× 100(1)
where V = total volume of the reduced NaOH (mL); c = concentration of the NaOH (mol/L); m = quantity of the sample (g); 282 = molar mass of oleic acid (g/mol).

#### 2.3.3. Peroxide Value (POV), Malondialdehyde (MDA) Content, and Carbonyl Group Value (CGV)

Peroxide value (meq O_2_/kg) analysis was performed according to the AOCS Official methods Ja 8–87 [22]. MDA content and CGV were determined in accordance with the National Standards of PR China GB/T 5009, 181-2016 [23] and GB/T 5009, 230-2016 [24], respectively.

### 2.4. Determination of the Contents of HAA Precursors

#### 2.4.1. Creatine Content

The creatine contents of bacon were determined based on the methods of our previous work [25]. The results were presented as mg/g of dry matter of fried and unfried bacon (control).

#### 2.4.2. Glucose Content

The glucose contents of bacon were quantified by reference to Zhang et al. [26] with some modifications. Five grams of fried bacon and unfried bacon (control) were mixed with double-distilled water (100 mL) and boiled for 1 min on an electric stove. Subsequently, the boiled solution was mixed with saturated lead acetate (2 mL) and cooled to 25 °C. The mixture was diluted to 250 mL with double distilled water and stood for 12 h. After that, 100 mL of the supernatant solution was transferred to a fresh tube and mixed with potassium oxalate (2.5 g). The mixture was centrifuged at 1200 g (10 min), and the supernatant (2 mL) was transferred to a fresh tube, mixed with 1 mL of phenol solution (5 g/L), and shaken thoroughly. Next, the concentrated sulfuric acid (5 mL) was added and incubated in a 25 °C water bath (15 min). Finally, the solution absorbance was detected at 490 nm. The data are presented as mg/g of dry matter of fried and unfried bacon (control).

#### 2.4.3. FAA Contents

The FAA contents of bacon were determined by reference to Du et al. [27]. The data are presented as mg/g of dry matter of fried and unfried bacon (control).

### 2.5. Extraction and Determination of HAAs in Bacon and Frying Oil

HAAs were extracted and analyzed in bacon following the procedure adopted by Du et al. [27]. Briefly, the fried and unfried sliced bacon (2 g), respectively, were chopped and added 200 μL (200 μg/L) TriMeIQx, and subsequently homogenized in 9.8 mL extraction solution (NaOH (40 g/L): methanol, 7:3). After that, the homogenizer probe was rinsed with 10 mL above extraction solution and the rinses were collected. The obtained mixture was centrifuged for 10 min (10,750× *g*). Then, the supernatant (10 mL) was subjected to Oasis MCX cartridge. Finally, the eluate from the cartridge was nitrogen-concentrated and then residue was dissolved in 1 mL mixed solution (acetate ammonium acetate buffer: acetonitrile, 1:1) and filtered into vials (Agilent Technologies). For the identification and quantification of HAAs, 2 μL sample was injected into the Shimadzu Nexera LC-30 (Shenyang, China) coupled to triple quadrupole mass spectrometry (UHPLC-MS/MS) equipped with Shim-pack XR-ODS reversed-phase analytical column (2.2 μm C18 III, 2.0 mm × 150 mm). The ammonium acetate (solvent A, 10 mmol/L) and acetonitrile (solvent B) were used for gradient elution and the analytical column was performed at 40 °C. The standards contained HAAs at 0.1, 0.5, 1.0, 2.0, 5.0, 10, 20, and 50 g/L. The residue of vegetable oil (5 mL) remaining after frying was directly added to the preconditioned cartridges, and the extraction and detection were the same as described above.

### 2.6. Sensory Evaluation

The panelists (*n* = 18) were selected from faculty and students of Northeast Agricultural University to evaluate the sensory quality of fried bacon. The panelists were trained for three preliminary sessions, and all the panelists assessed the samples across sessions. Three attributes (color, flavor, and taste) were evaluated [2]. Sensory scores for fried bacon were rated on a 7-point descriptive scale: color (1 = light; 7 = dark), flavor (1 = mild; 7 = intense), and taste (1 = mild; 7 = intense). The bacon was fried, cut into slices (1 cm × 3 cm), and served immediately. There were three sensory sessions. For each session, five packages of bacon were used for sensory evaluation.

### 2.7. Statistical Analysis

All related experiments assays were performed in triplicate and three independent batches of bacon (replicates) were manufactured. Data obtained from the statistical analysis were calculated using Statistix 8.1 (St. Paul, MN, USA) by Tukey’s multiple comparison for analysis of variance. The data were presented as the mean ± standard error (SE). The statistical significance was set at *p* < 0.05.

## 3. Results and Discussion

### 3.1. Fatty Acid Composition and Quality Characteristics of Vegetable Oils

The fatty acid composition of different vegetable oils is listed in Table 1. The content of total saturated fatty acids (SFAs) in rapeseed oil was lower than that of the other vegetable oils (*p* < 0.05). Palmitic acid and stearic acid were the main SFAs in all vegetable oils, consistent with previous reports [16]. Rapeseed oil recorded the lowest level of palmitic acid (*p* < 0.05), and there was no significant difference in the contents of palmitic acid between peanut oil, corn oil, and soybean oil (*p* > 0.05). Hu et al. [28] suggested that a relatively high intake of palmitic acid in the daily diet might increase the risk of coronary heart disease.

Peanut oil showed the highest contents of behenic acid and lignoceric acid among the vegetable oils (*p* < 0.05) and this contributed to the relatively higher level of SFAs in peanut oil (*p* < 0.05). Contrary to the SFAs, the highest levels of total monounsaturated fatty acids (MUFAs) and oleic acid were determined in rapeseed oil (*p* < 0.05). Although rapeseed oil displayed the lowest content of polyunsaturated fatty acids (PUFAs) and linoleic acid (*p* < 0.05), it had the highest content of α-linolenic acid *(p* < 0.05).

Besides the fatty acid composition, the vegetable oils were also evaluated for their FFA contents, POV, MDA content, and CGV (Table 1). All these individual parameters can reflect the complex chemical composition and quality characteristics of vegetable oils. The FFA contents of peanut oil, corn oil, rapeseed oil, sunflower seed oil, and soybean oil were 0.29, 0.10, 0.06, 0.09, and 0.11%, respectively. These data were higher than that of the reported by Li et al. [29] for 18 different vegetable oils (almond, blend 1–8, camellia, corn, palm, peanut, rapeseed, sesame, soybean, sunflower, and zanthoxylum oil), which showed FFA contents ranging from 0.10 to 0.13 mg/g. One reason for this difference might be that pigment levels of oils can change during storage and the spectral absorbance might also be affected by different oil colors [29]. The POV contents of all vegetable oils ranged from 0.07 to 0.23 g/100 g. All vegetable oil samples had a POV within the legal limit of 0.25 meq O_2_/kg [30]. The highest content of POV was found in sunflower seed oil (*p* < 0.05) and peanut oil had the highest content of MDA (*p* < 0.05). Oils with a lower level of unsaturated triacylglycerols show better oxidative stability [16,31], which agrees with our observation (Table 1). Additionally, the CGV of five vegetable oils ranked as soybean oil > sunflower seed oil > rapeseed oil > corn oil > peanut oil (*p* < 0.05).

### 3.2. The Contents of HAA Precursors in Bacon

Creatine is an important precursor of HAAs in meat samples [4]. Generally, creatine is required to form creatinine, which combines with aldehydes and pyridine or pyrazine to form polar HAAs [32]. The creatine contents of bacon fried in five different vegetable oils are shown in Figure 1. The creatine content was higher in unfried bacon (control) than in fried bacon (*p* < 0.05), which suggests that creatine was converted to HAAs during frying [2]. Regarding the type of frying oil, the level of creatine was lowest in bacon fried in rapeseed oil (*p* < 0.05), and the corn oil, sunflower seed oil, and soybean oil did not affect the level of creatine in fried samples (*p* > 0.05). This indicates that rapeseed oil may be more likely than the other oils to convert the creatine in bacon into HAAs, especially polar HAAs, during frying. Evidence for this notion is the relatively higher levels of polar HAAs in rapeseed oil-fried bacon as presented in Table 3.

Similar to creatine, reducing sugars such as glucose are also important precursors of HAAs. The difference is that glucose not only participates in the formation of polar HAAs but also promotes the formation of non-polar HAAs [32]. Figure 1 shows that the glucose contents were higher in unfried bacon (control) than in fried bacon (*p* < 0.05). Bacon fried in corn oil had the lowest glucose content among the frying treatments (*p* < 0.05) and rapeseed oil and soybean oil were not affected by the glucose content in fried bacon samples (*p* > 0.05). The glucose in bacon may combine with other precursors, such as amino acids, through the aldol reaction during frying to produce intermediates, such as glucose tryptophan Amadori products, which may further convert into HAAs [4]. Similarly, Gibis et al. [33] found that frying decreased the glucose content of beef patties.

FAAs are also important precursors of HAAs. The contents of FAAs in unfried and fried bacon are listed in Table 2. The total FAA contents of unfried bacon (control) were higher than those of the fried bacon samples (*p* < 0.05), perhaps indicating the involvement of FAAs in the formation of HAAs. Pyridine and pyrazine can be formed from some FAAs that act as their nitrogen source during frying [26]. Several reports showed that glycine, lysine, threonine, alanine, and serine were related to the formation of MeIQx [34], lysine, aspartic, tyrosine, isoleucine, and phenylalanine were responsible for the formation of PhIP [34], alanine, threonine, and lysine have some connection with the development of 4,8-DiMeIQx [4], and tryptophan led to the generation of β-carbolines [35].

The total FAA contents of bacon were lower after frying in corn oil than in rapeseed oil (*p* < 0.05). Peanut oil, sunflower seed oil, and soybean oil were not affected by the contents of FAA in fried samples (*p* > 0.05). Therefore, corn oil might be more likely to promote the conversion of FAAs into HAAs during frying, as demonstrated by the evidence that bacon fried in corn oil had the highest level of total HAAs among the frying treatments (Table 3).

### 3.3. HAA Contents

The contents of HAAs in fried bacon and vegetable oil after frying are listed in Table 3 and Table 4, respectively. Eight of the twelve HAAs analyzed, including polar and non-polar HAAs, were detected in bacon. Unfried bacon (control) had a lower content of total HAAs than the bacon samples fried in different vegetable oils (*p* < 0.05). Various precursors, such as creatine, glucose, and FAAs, are converted into HAAs during the frying process [2]. Ekiz et al. [16] found noticeably higher HAA contents in deep-fried meatballs compared to raw meatballs. Likewise, Johansson et al. [36] detected remarkedly higher HAA contents in a model system with corn oil or olive oil compared to the control. MUFAs and PUFAs in vegetable oils may be oxidized and decomposed to produce free radicals, aldehydes, and ketones during the heating process [31]. Smoked bacon contains substantial amounts of organic compounds, such as aldehydes, alcohols, ketones, and furans [18], and some of these compounds can serve as intermediates or precursors, such as Strecker aldehydes, pyrazines, and pyridines, in the Maillard reaction to form HAAs [26].

**Table 3 foods-11-03491-t003:** Influence of the different vegetable oils on the HAA concentration (ng/g) in pan-fried bacon.

	Control	B–PO	B–CO	B–RO	B–SSO	B–SO
Norharman	5.40 ± 0.33 ^E^	16.17 ± 1.12 ^D^	45.66 ± 2.61 ^A^	21.10 ± 1.28 ^CD^	25.33 ± 0.66 ^C^	35.73 ± 1.85 ^B^
Harman	0.19 ± 0.01 ^E^	6.23 ± 0.43 ^D^	20.83 ± 1.19 ^A^	10.32 ± 0.60 ^C^	8.49 ± 0.22 ^CD^	14.40 ± 0.75 ^B^
AαC	1.66 ± 0.14 ^A^	0.38 ± 0.03 ^B^	0.29 ± 0.03 ^B^	0.29 ± 0.02 ^B^	0.27 ± 0.01 ^B^	0.42 ± 0.02 ^B^
MeAαC	n.d.	7.99 ± 0.55 ^B^	18.47 ± 1.06 ^A^	8.15 ± 0.47 ^B^	5.48 ± 0.14 ^B^	8.05 ± 0.42 ^B^
**Non-polar HAAs**	**7.25 ± 0.49 ^D^**	**30.77 ± 2.13 ^C^**	**85.26 ± 4.88 ^A^**	**39.86 ± 2.30 ^C^**	**39.57 ± 1.03 ^C^**	**58.60 ± 3.03 ^B^**
IQ	n.d.	n.d.	n.d.	n.d.	n.d.	n.d.
7,8-DiMeIQx	n.d.	n.d.	n.d.	n.d.	n.d.	n.d.
MeIQx	n.d.	n.d.	n.d.	n.d.	n.d.	n.d.
IQx	0.39 ± 0.02 ^D^	0.96 ± 0.07 ^A^	0.89 ± 0.09 ^A^	0.55 ± 0.03 ^C^	0.54 ± 0.01 ^C^	0.68 ± 0.04 ^B^
4,8-DiMeIQx	0.51 ± 0.03 ^B^	0.76 ± 0.04 ^A^	0.71 ± 0.04 ^A^	0.53 ± 0.03 ^B^	0.50 ± 0.02 ^B^	0.59 ± 0.04 ^B^
MeIQ	0.64 ± 0.03 ^C^	0.70 ± 0.05 ^C^	1.25 ± 0.13 ^B^	0.85 ± 0.05 ^C^	1.10 ± 0.03 ^B^	1.84 ± 0.09 ^A^
PhIP	1.99 ± 0.14 ^D^	5.31 ± 0.37 ^C^	7.05 ± 0.40 ^B^	9.73 ± 0.51 ^A^	3.11 ± 0.08 ^D^	2.66 ± 0.37 ^D^
Phe-P-I	n.d.	n.d.	n.d.	n.d.	n.d.	n.d.
**Polar HAAs**	**3.52 ± 0.18 ^E^**	**7.73 ± 0.51 ^C^**	**9.91 ± 0.94 ^B^**	**11.66 ± 0.67 ^A^**	**5.52 ± 0.12 ^D^**	**5.77 ± 0.30 ^D^**
**Total HAAs**	**10.77 ± 0.66 ^D^**	**38.50 ± 2.65 ^C^**	**95.17 ± 5.40 ^A^**	**51.52 ± 2.98 ^BC^**	**44.82 ± 1.14 ^C^**	**64.37 ± 3.32 ^B^**

Different superscript letters (^A–E^) within the same row differ significantly (*p* < 0.05). Data were presented as mean ± standard errors. n.d., the concentration is below the detection limit. Control, unfried bacon; B–PO, bacon fried in peanut oil; B–CO, bacon fried in corn oil; B–RO, bacon fried in rapeseed oil; B–SSO, bacon fried in sunflower seed oil; B–SO, bacon fried in soybean oil.

The total HAA contents of fried bacon varied between 38.50 and 95.17 ng/g. These values were higher than those detected in fried meatballs [16], fried beef burgers [3], and roasted beef patties [17]. Some of the many factors that influence HAA formation are the type and number of HAAs detected, cooking method, cooking temperature and time, and meat type [12], and, as shown in the current study, the type of frying oil.

Bacon fried in corn oil had the highest content of HAAs among the frying treatments (*p* < 0.05), and peanut oil, rapeseed oil, and sunflower seed oil were not affected by the HAA contents in fried bacon samples (*p* > 0.05). This marked impact of vegetable oil on the formation of HAA in fried bacon is similar to a previous study that showed different oils differentially affected the contents of HAAs in roasted beef patties [17]. Regarding specific oils, rapeseed oil resulted in a more deleterious reaction outcome than sunflower seed oil regarding the total contents of HAAs in fried meatballs [16] and beef burgers [3]. However, only polar HAAs were detected in those reports. If we consider only polar HAAs in our study, it also supports that frying in rapeseed oil leads to more HAAs in bacon than frying in sunflower seed oil (*p* < 0.05) (Table 3).

As shown in Table 4, frying significantly affected the HAA contents of the oils. The total HAAs after frying were highest in peanut oil (*p* < 0.05) and lowest in rapeseed oil (27.07 ng/g) and soybean oil (20.95 ng/g). Corn oil and sunflower seed oil had comparable contents (*p* < 0.05). In addition, the total HAA contents of the oil after frying differed from that of the fried bacon. The content was higher in peanut oil (117.11 ng/g) than in bacon fried in peanut oil (38.50 ng/g) (*p* < 0.05). On the contrary, corn oil (43.47 ng/g), rapeseed oil (27.07 ng/g), and soybean oil (20.95 ng/g) all had lower contents than those found in bacon fried in corn oil (95.17 ng/g), rapeseed oil (51.52 ng/g), and soybean oil (64.37 ng/g) (*p* < 0.05), respectively. This difference could be attributed to the concentrations of total FFAs and MDA, which were higher in peanut oil than in the other vegetable oils (Table 1). FFAs are easily oxidized and decomposed to generate free radicals from hydroperoxides during frying, which, in turn, further promotes the formation of HAAs and other harmful substances, including polycyclic aromatic hydrocarbons, acrolein, and other carcinogens [37].

When the vegetable oils and bacon samples were assessed separately after frying, the contents of non-polar HAA (Norharman, Harman, AαC, and MeAαC) and total HAAs had similar trends (Table 3 and Table 4). In addition, the non-polar HAA contents were higher than the polar HAA (IQx, 4,8-DiMeIQx, MeIQ, and PhIP) contents of all samples (*p* < 0.05). This finding is similar to that observed in beef burgers fried in virgin olive oil and rapeseed oil, which revealed that non-polar HAAs were principally accumulated during the frying process [38].

Norharman had the highest content among the non-polar HAAs in each treatment. Bacon fried in peanut oil had higher Norharman and Harman contents than that of other fried bacon samples (*p* < 0.05), and Norharman and Harman contents were comparable between bacon fried in rapeseed oil and sunflower seed oil. Moreover, Norharman and Harman contents were highest in the remaining peanut oil (*p* < 0.05). Both compounds were present at higher levels than those detected by Li et al. [17] in roast beef patties fried in 10 different vegetable oils. Liu et al. [39] demonstrated that Harman and Norharman were more degraded during heating (150 and 180 °C) in MUFA-type (oleic acid-rich) oil blends than PUFA-rich oil blends, and oxidative degradation was speculated as the likely mechanism. Likewise, Randel et al. [40] showed that Norharman and Harman were degraded by more than 40% in rapeseed oil (130 and 180 °C) but by only 20–30% in sunflower seed oil and corn oil. Our results also confirmed that the Norharman and Harman contents of remaining corn oil (of which 24.54% is MUFA) and sunflower seed oil (of which 27.20% is MUFA) were higher than that of rapeseed oil (of which 58.64% is MUFA) (*p* < 0.05).

The concentrations of PhIP and total polar HAAs (4,8-DiMeIQx, MeIQ, IQx, and PhIP) in bacon fried in different vegetable oils were higher than those in unfried bacon (*p* < 0.05). Bacon fried in rapeseed oil had the highest PhIP and total polar HAA contents among the fried bacon samples (*p* < 0.05), and both amounts were comparable between bacon fried in sunflower seed oil and soybean oil (*p* > 0.05). The lowest contents of creatine and FAAs in bacon fried in rapeseed oil may be the reason for its higher polar HAA and PhIP contents (Figure 1 and Table 2). In addition, the statistically highest polar HAA contents of remaining oil were found in corn oil (4.55 ng/g) and peanut oil (4.28 ng/g), followed by sunflower seed oil (3.43 ng/g) > rapeseed oil (2.66 ng/g) > soybean oil (1.71 ng/g). The remaining corn oil had the highest PhIP content among the vegetable oils (*p* < 0.05). As Randel et al. [40] concluded, the degradation of HAA in frying fats is closely correlated to the type of frying fat and is promoted by lipid oxidation products. They reported that the degradation rates of IQx, 4,8-DiMeIQx, and PhIP in sunflower seed oil were higher than those in corn oil during heating.

### 3.4. Sensory Evaluation

Table 5 shows the results of the descriptive sensory analysis. Bacon fried in peanut oil and corn oil had higher color scores than the sample fried in soybean oil (*p* < 0.05), which, at least in part, could be explained by the relatively darker appearance of peanut oil and corn oil (Figure 2). The color scores were comparable among the samples fried in rapeseed oil, sunflower seed oil, and soybean oil (*p* > 0.05). Unlike the color, the flavor and taste scores of fried bacon were not affected by the vegetable oil types (*p* > 0.05). This may be due to the fact that the vegetable oils are refined and the flavor compounds of the vegetable oils have little difference, so there were no significant differences in the flavor and taste scores between the samples.

## 4. Conclusions

This study evaluated the physicochemical characteristics of five vegetable oils and their effect on the HAA contents in fried bacon and the oils after frying. Frying significantly increased the total HAA contents and decreased the precursor contents (FAAs, creatine, and glucose) of bacon. Peanut oil, rapeseed oil, and sunflower seed oil had the weakest promoting effect on the total HAAs of fried bacon, and rapeseed and soybean oils had the lowest content of total HAAs among the vegetable oils after frying. Moreover, the vegetable oil type did not impact the flavor and taste of fried bacon. Hence, the vegetable oil types significantly influenced the HAA contents of fried bacon and peanut oil was the most suitable frying oil in terms of HAA content.

## Figures and Tables

**Figure 1 foods-11-03491-f001:**
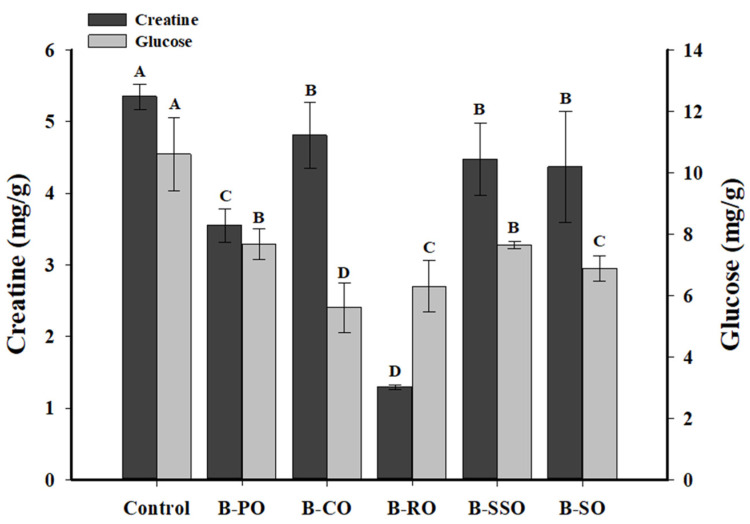
Chang of creatine and glucose in fried bacon using different oils. Significant differences (*p* < 0.05) within same index are represented by different uppercase letters (A–D). Control, unfried bacon; B–PO, bacon fried in peanut oil; B–CO, bacon fried in corn oil; B–RO, bacon fried in rapeseed oil; B–SSO, bacon fried in sunflower seed oil; B–SO, bacon fried in soybean oil.

**Figure 2 foods-11-03491-f002:**
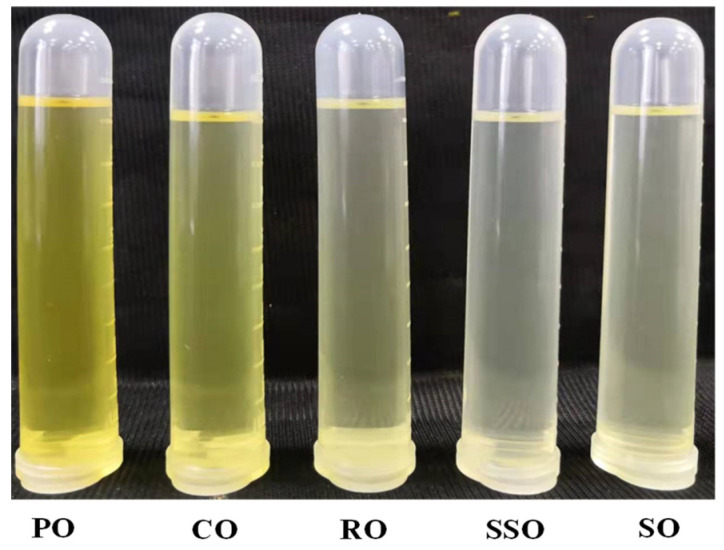
Appearance of the different vegetable oils before frying. PO, peanut oil; CO, corn oil; RO, rapeseed oil; SSO, sunflower seed oil; SO, soybean oil.

**Table 1 foods-11-03491-t001:** Fatty acid composition (%), free fatty acid (%), peroxide value (meq O_2_/kg), malondialdehyde (mg/kg), and carbonyl group value (meq/kg) of the different vegetable oils.

	PO	CO	RO	SSO	SO
Tetradecanoic acid	0.02 ± 0.00 ^C^	0.02 ± 0.00 ^C^	0.04 ± 0.00 ^B^	0.06 ± 0.00 ^A^	0.07 ± 0.00 ^A^
Pentadecanoic acid	0.01 ± 0.00 ^C^	n.d.	0.02 ± 0.00 ^A^	0.01 ± 0.00 ^B^	0.01 ± 0.00 ^B^
Hexadecanoic acid	10.38 ± 0.42 ^A^	11.34 ± 0.33 ^A^	4.49 ± 0.16 ^C^	6.70 ± 0.31 ^B^	10.77 ± 0.31 ^A^
Heptadecanoic acide	0.06 ± 0.00 ^B^	0.06 ± 0.00 ^B^	0.05 ± 0.00 ^C^	0.03 ± 0.00 ^D^	0.09 ± 0.01 ^A^
Stearic acid	3.23 ± 0.13 ^C^	1.78 ± 0.05 ^D^	1.81 ± 0.06 ^D^	3.88 ± 0.18 ^B^	4.54 ± 0.13 ^A^
Arachidic acid	1.50 ± 0.06 ^A^	0.42 ± 0.01 ^C^	0.67 ± 0.02 ^B^	0.27 ± 0.01 ^D^	0.44 ± 0.01 ^C^
Heneicosanoic acid	0.01 ± 0.00 ^C^	n.d.	0.02 ± 0.00 ^B^	n.d.	0.03 ± 0.00 ^A^
Behenic acid	3.40 ± 0.13 ^A^	0.12 ± 0.01 ^D^	0.34 ± 0.01 ^C^	0.82 ± 0.04 ^B^	0.45 ± 0.01 ^C^
Tricosanoic acid	0.03 ± 0.00 ^B^	0.01 ± 0.00 ^D^	0.02 ± 0.00 ^C^	0.03 ± 0.00 ^B^	0.04 ± 0.00 ^A^
Lignoceric acid	1.55 ± 0.06 ^A^	0.15 ± 0.01 ^B^	0.14 ± 0.01 ^B^	0.27 ± 0.01 ^C^	0.15 ± 0.01 ^B^
∑**SFA**	**20.20 ± 0.82 ^A^**	**13.90 ± 0.40 ^C^**	**7.60 ± 0.26 ^E^**	**12.06 ± 0.56 ^D^**	**16.56 ± 0.48 ^B^**
Cis-9-hexadecenoic acid	0.05 ± 0.00 ^D^	0.07 ± 0.00 ^C^	0.20 ± 0.01 ^A^	0.10 ± 0.01 ^B^	0.07 ± 0.00 ^C^
Cis-9-Octadecenic acid	39.78 ± 1.61 ^B^	24.22 ± 0.70 ^C^	56.88 ± 1.97 ^A^	26.98 ± 1.25 ^C^	23.03 ± 0.66 ^C^
Cis-11-eicosenoic acid	1.24 ± 0.05 ^A^	0.25 ± 0.01 ^B^	1.27 ± 0.04 ^A^	0.13 ± 0.01 ^C^	0.27 ± 0.01 ^B^
Cis-13-docosaenoic acid	0.09 ± 0.00 ^B^	n.d.	0.16 ± 0.01 ^A^	n.d.	n.d.
Cis-15-catecosenoic acid	n.d.	n.d.	0.14 ± 0.01 ^A^	n.d.	n.d.
∑**MUFA**	**41.16 ± 1.66 ^B^**	**24.54 ± 0.71 ^C^**	**58.64 ± 2.03 ^A^**	**27.20 ± 1.26 ^C^**	**23.37 ± 0.67 ^C^**
Cis-9,12-octadecadienoic acid	36.46 ± 1.47 ^B^	53.86 ± 1.55 ^A^	18.97 ± 0.66 ^C^	56.32 ± 2.60 ^A^	49.59 ± 1.43 ^A^
Cis-9,12,15-octadecatrienoic acid	0.08 ± 0.01 ^C^	0.80 ± 0.04 ^B^	5.74 ± 0.34 ^A^	0.09 ± 0.01 ^C^	6.07 ± 0.30 ^A^
Cis-11,14-eicosadienoic acid	0.02 ± 0.00 ^B^	0.02 ± 0.00 ^B^	0.05 ± 0.00 ^A^	n.d.	0.03 ± 0.00 ^B^
∑**PUFA**	**36.56 ± 1.48 ^B^**	**54.69 ± 1.58 ^A^**	**24.76 ± 0.86 ^C^**	**56.41 ± 2.61 ^A^**	**55.69 ± 1.61 ^A^**
∑**MUFA/**∑**SFA**	2.04 ± 0.08 ^BC^	1.77 ± 0.05 ^CD^	7.71 ± 0.27 ^A^	2.26 ± 0.10 ^B^	1.41 ± 0.07 ^D^
∑**PUFA/**∑**SFA**	1.81 ± 0.07 ^D^	3.93 ± 0.11 ^B^	3.26 ± 0.11 ^C^	4.68 ± 0.22 ^A^	3.36 ± 0.10 ^C^
Free fatty acid	0.29 ± 0.01 ^A^	0.10 ± 0.01 ^BC^	0.06 ± 0.01 ^D^	0.09 ± 0.02 ^C^	0.11 ± 0.01 ^B^
Peroxide value	0.19 ± 0.02 ^B^	0.12 ± 0.01 ^C^	0.07 ± 0.01 ^D^	0.23 ± 0.02 ^A^	0.16 ± 0.01 ^B^
Malondialdehyde	0.41 ± 0.05 ^A^	0.28 ± 0.02 ^BC^	0.34 ± 0.03 ^B^	0.25 ± 0.02 ^C^	0.31 ± 0.01 ^BC^
Carbonyl group value	0.42 ± 0.06 ^E^	0.86 ± 0.04 ^D^	1.23 ± 0.06 ^C^	1.97 ± 0.07 ^B^	2.26 ± 0.18 ^A^

Different superscript letters (^A–E^) within the same row differ significantly (*p* < 0.05). Data were presented as mean ± standard errors. n.d., the concentration is below the detection limit. PO, peanut oil; RO, rapeseed oil; CO, corn oil; SO, soybean oil; SSO, sunflower seed oil. ∑SFA, ∑MUFA, ∑PUFA are the total saturated fatty acid, total monounsaturated fatty acid, and total polyunsaturated fatty acids, respectively.

**Table 2 foods-11-03491-t002:** Chang of FAAs (ug/g) in fried bacon using different vegetable oils.

	Control	B–PO	B–CO	B–RO	B–SSO	B–SO
Asp	0.29 ± 0.02 ^A^	0.17 ± 0.02 ^C^	0.10 ± 0.01 ^D^	0.20 ± 0.02 ^B^	0.20 ± 0.01 ^B^	0.16 ± 0.01 ^C^
Glu	1.42 ± 0.08 ^A^	1.44 ± 0.05 ^A^	1.42 ± 0.04 ^A^	1.45 ± 0.03 ^A^	1.47 ± 0.03 ^A^	1.42 ± 0.04 ^A^
Ser	0.19 ± 0.01 ^D^	0.30 ± 0.01 ^B^	0.22 ± 0.01 ^C^	0.32 ± 0.01 ^B^	0.47 ± 0.02 ^A^	0.30 ± 0.01 ^B^
Gly	1.19 ± 0.07 ^A^	0.21 ± 0.01 ^B^	0.12 ± 0.01 ^C^	0.22 ± 0.02 ^B^	0.25 ± 0.01 ^B^	0.21 ± 0.01 ^B^
His	1.32 ± 0.08 ^A^	1.03 ± 0.04 ^B^	0.78 ± 0.02 ^C^	1.11 ± 0.03 ^B^	0.84 ± 0.02 ^C^	0.99 ± 0.03 ^B^
Thr	1.34 ± 0.08 ^A^	0.74 ± 0.03 ^BC^	0.53 ± 0.02 ^D^	0.80 ± 0.02 ^B^	0.72 ± 0.02 ^BC^	0.63 ± 0.02 ^CD^
Ala	2.03 ± 0.06 ^A^	1.36 ± 0.05 ^BC^	0.66 ± 0.02 ^D^	1.48 ± 0.03 ^B^	1.34 ± 0.03 ^BC^	1.21 ± 0.03 ^C^
Arg	3.24 ± 0.19 ^A^	2.39 ± 0.08 ^B^	2.68 ± 0.08 ^B^	2.65 ± 0.06 ^B^	2.08 ± 0.05 ^C^	1.74 ± 0.05 ^D^
Pro	0.16 ± 0.01 ^A^	0.13 ± 0.01 ^C^	0.12 ± 0.01 ^D^	0.16 ± 0.01 ^A^	0.14 ± 0.01 ^B^	n.d.
Tyr	0.11 ± 0.01 ^AB^	0.12 ± 0.01 ^A^	0.12 ± 0.01 ^A^	0.12 ± 0.01 ^A^	0.10 ± 0.01 ^B^	0.11 ± 0.01 ^AB^
Val	0.11 ± 0.01 ^C^	0.21 ± 0.02 ^B^	0.23 ± 0.02 ^B^	0.25 ± 0.01 ^A^	0.22 ± 0.01 ^B^	0.21 ± 0.01 ^B^
Met	0.02 ± 0.01 ^E^	0.09 ± 0.01 ^C^	0.07 ± 0.01 ^D^	0.27 ± 0.01 ^B^	0.28 ± 0.01 ^B^	0.30 ± 0.01 ^A^
Cys	0.68 ± 0.04 ^BC^	0.72 ± 0.03 ^AB^	0.46 ± 0.01 ^D^	0.65 ± 0.02 ^BC^	0.61 ± 0.01 ^C^	0.76 ± 0.02 ^A^
Ile	0.39 ± 0.02 ^B^	0.46 ± 0.02 ^A^	0.42 ± 0.01 ^B^	0.41 ± 0.01 ^B^	0.39 ± 0.01 ^B^	0.40 ± 0.01 ^B^
Leu	0.09 ± 0.01 ^E^	0.21 ± 0.02 ^A^	0.16 ± 0.01 ^CD^	0.19 ± 0.01 ^B^	0.17 ± 0.01 ^C^	0.15 ± 0.01 ^D^
Phe	0.27 ± 0.02 ^A^	0.16 ± 0.01 ^B^	0.16 ± 0.02 ^B^	0.16 ± 0.01 ^B^	0.14 ± 0.01 ^B^	0.13 ± 0.01 ^B^
Lys	1.66 ± 0.10 ^A^	1.68 ± 0.06 ^A^	1.60 ± 0.08 ^A^	1.65 ± 0.07 ^A^	1.65 ± 0.07 ^A^	1.69 ± 0.04 ^A^
**Total FAA**	**14.50 ± 0.83 ^A^**	**11.44 ± 0.40 ^BC^**	**9.84 ± 0.28 ^C^**	**12.10 ± 0.28 ^B^**	**11.09 ± 0.26 ^BC^**	**10.44 ± 0.27 ^BC^**

Different superscript letters (^A–E^) within the same row differ significantly (*p* < 0.05). Data were presented as mean ± standard errors. n.d., the concentration is below the detection limit. Control, unfried bacon; B–PO, bacon fried in peanut oil; B–CO, bacon fried in corn oil; B–RO, bacon fried in rapeseed oil; B–SSO, bacon fried in sunflower seed oil; B–SO, bacon fried in soybean oil.

**Table 4 foods-11-03491-t004:** The HAA content (ng/g) in different vegetable oil after frying.

	B–PO	B–CO	B–RO	B–SSO	B–SO
Norharman	77.15 ± 4.21 ^A^	23.74 ± 1.64 ^C^	12.82 ± 0.74 ^D^	31.86 ± 0.88 ^B^	12.57 ± 0.72 ^D^
Harman	28.00 ± 1.53 ^A^	8.52 ± 0.59 ^B^	5.88 ± 0.34 ^C^	9.98 ± 0.28 ^B^	4.55 ± 0.26 ^C^
AαC	0.52 ± 0.03 ^B^	0.48 ± 0.03 ^B^	0.14 ± 0.01 ^D^	0.87 ± 0.02 ^A^	0.33 ± 0.02 ^C^
MeAαC	7.17 ± 0.39 ^A^	6.17 ± 0.43 ^B^	5.57 ± 0.32 ^B^	5.58 ± 0.16 ^B^	1.79 ± 0.10 ^C^
**Non-polar HAAs**	**112.83 ± 6.16 ^A^**	**38.92 ± 2.70 ^B^**	**24.41 ± 1.41 ^C^**	**48.29 ± 1.34 ^B^**	**19.24 ± 1.10 ^C^**
IQ	n.d.	n.d.	n.d.	n.d.	n.d.
7,8-DiMeIQx	n.d.	n.d.	n.d.	n.d.	n.d.
MeIQx	n.d.	n.d.	n.d.	n.d.	n.d.
IQx	0.79 ± 0.04 ^A^	0.52 ± 0.03 ^B^	0.77 ± 0.05 ^A^	0.48 ± 0.01 ^B^	0.26 ± 0.02 ^C^
4,8-DiMeIQx	0.66 ± 0.05 ^A^	0.56 ± 0.04 ^A^	0.43 ± 0.03 ^B^	0.30 ± 0.01 ^C^	0.44 ± 0.04 ^B^
MeIQ	0.22 ± 0.01 ^D^	0.39 ± 0.03 ^B^	0.31 ± 0.02 ^C^	1.31 ± 0.03 ^A^	0.26 ± 0.02 ^CD^
PhIP	2.60 ± 0.14 ^B^	3.08 ± 0.21 ^A^	1.16 ± 0.07 ^C^	1.34 ± 0.03 ^C^	0.74 ± 0.04 ^D^
Phe-P-I	n.d.	n.d.	n.d.	n.d.	n.d.
**Polar HAAs**	**4.28 ± 0.16 ^A^**	**4.55 ± 0.31 ^A^**	**2.66 ± 0.16 ^C^**	**3.43 ± 0.08 ^B^**	**1.71 ± 0.06 ^D^**
**Total HAAs**	**117.11 ± 6.32 ^A^**	**43.47 ± 3.01 ^B^**	**27.07 ± 1.57 ^C^**	**51.72 ± 1.42 ^B^**	**20.95 ± 1.15 ^C^**

Different superscript letters (^A–D^) within the same row differ significantly (*p* < 0.05). Data were presented as mean ± standard errors. n.d., the concentration is below the detection limit. PO, peanut oil; CO, corn oil; RO, rapeseed oil; SFO, sunflower seed oil; SO, soybean oil.

**Table 5 foods-11-03491-t005:** Influence of different vegetable oils on sensory quality of fried bacon.

	B–PO	B–CO	B–RO	B–SSO	B–SO
Color	5.41 ± 0.23 ^A^	5.60 ± 0.19 ^A^	5.01 ± 0.16 ^AB^	5.00 ± 0.28 ^AB^	4.46 ± 0.24 ^B^
Flavor	4.91 ± 0.18 ^A^	5.28 ± 0.18 ^A^	4.97 ± 0.17 ^A^	5.43 ± 0.20 ^A^	5.00 ± 0.20 ^A^
Taste	5.38 ± 0.14 ^A^	5.36 ± 0.17 ^A^	5.20 ± 0.17 ^A^	5.38 ± 0.16 ^A^	5.21 ± 0.20 ^A^

Different superscript letters (^A^,^B^) within the same row differ significantly (*p* < 0.05). Data were presented as mean ± standard errors. B–PO: bacon fried in peanut oil; B–CO: bacon fried in corn oil; B–RO: bacon fried in rapeseed oil; B–SSO: bacon fried in sunflower seed oil; B–SO: bacon fried in soybean oil.

## Data Availability

The data presented in this study are available in the article.

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
