# Peer review of "Understanding the Development of Heterocyclic Aromatic Amines in Fried Bacon and in the Remaining Oil after Pan-Frying in Five Different Vegetable Oils"

_foods, 2022, doi:10.3390/foods11213491_

Round 1
Reviewer 1 Report
The manuscript is presenting the effect of five different vegetable oils (peanut, corn, rapeseed, sunflower seed, and soybean) on the development of HAAs in fried bacon as well as in the remaining oil. The level of FAAs, creatine, and glucose and the sensory qualities were assessed too.
The individual chapters provide information on HAAs and the possibility of their formation in food of animal origin during its high-temperature thermal processing.
The abstract of the reviewed paper has informative and fully covered the content of the manuscript. The length of the manuscript is appropriate in relation to the content.
The article has been prepared in accordance with the instructions for authors. English is understandable. This article requires some minor clarifications. After reading through the manuscript, I found several issues that should be addressed by the authors:
In the reviewer’s opinion the objective of the study is correctly formulated and sufficiently described. The experimental design and methods are appropriate for the purposes of the study, and investigations has been conducted in an ethically acceptable manner. The abstract of the reviewed paper has informative and fully covered the content of the manuscript. The length of the manuscript is appropriate in relation to the content.
In the reviewer’s opinion the objective of the study is basically correctly formulated and described. The experimental design and methods are appropriate for the purposes of the study, and investigations has been conducted in an ethically acceptable manner.
However, the questions about the research material do arise:
1. Were the vegetable oils used to the thermal processing of bacon was refined or expeller pressing?
2. The description of the preparation of the raw meat material (pork belly) for further stages of the research causes confusion:
- the statement “stabilized” (Line 85) should be understood as “curing”? and next: “… randomly divided into three equal groups” (Line 86). What did the authors mean?
- what material (-s) are the bags made of? How was vacuum pressure value in bag (expressed in % and kPa)? What was the permeability of the packaging film in relation to O2 and moisture vapor (expressed in x cm3´m2´24h´x MPa)? – Line 90.
- Line 91 – What method was used to freeze the sample? What was the freezing time? What method was used to thaw the samples? What was the thawing time?
- Line 95 “pan” …. what pan …. frying pan (non-stick-pan), grill pan? What was the plate of the pan?
Reviewer comments to Authors:
Line 14 and next – is P<0.05, should be P0.05, if differences between means were statistically significant.
Table 1. – pp. 5-6, Please standardize the nomenclature of fatty acids, i.e. use trivial or systematic names.
Reviewer 2 Report
In my opinion, this article is very interesting, the authors analyzed physicochemical characteristics of five vegetable oils and their effect on the HAA contents in fried bacon and the oils after frying.
Dear authors, attached my few suggestion for improvement of the article:
Pag 2 line 58: The physicochemical properties of five vegetable oils, peanut, corn, rapeseed, sunflower seed, and soybean were studied. Why the authors did not consider extra virgin olive oil in the experimental plan? In the objectives of this article, the authors should explain better why they have chosen the five vegetable oils.
Pag 4 line 177: In the sensory analysis how many samples were used? Is there any replication conducted for sensory evaluation?
Pag 10 line 378: Elaborate more on the sensory analysis findings.
